# Plasma-Activated Media Produced by a Microwave-Excited Atmospheric Pressure Plasma Jet Is Effective against Cisplatin-Resistant Human Bladder Cancer Cells In Vitro

**DOI:** 10.3390/ijms25021249

**Published:** 2024-01-19

**Authors:** Ara Jo, Hea-Min Joh, Jin-Hee Bae, Sun-Ja Kim, Jin-Woong Chung, Tae-Hun Chung

**Affiliations:** 1Department of Biological Sciences, Dong-A University, Busan 49315, Republic of Korea; dkfk2111@naver.com; 2Department of Physics, Dong-A University, Busan 49315, Republic of Koreajinnybae@kaist.ac.kr (J.-H.B.); sjkim@dau.ac.kr (S.-J.K.)

**Keywords:** microwave-excited atmospheric pressure plasma jet, plasma-activated medium-cell interaction, reactive oxygen and nitrogen species, cell viability, cisplatin-resistant human bladder cancer cells, *HER* expression

## Abstract

Media exposed to atmospheric pressure plasma (APP) produce reactive oxygen and nitrogen species (RONS), with hydrogen peroxide (H_2_O_2_), nitrite (NO_2_^−^), and nitrate (NO_3_^−^) being among the most detected species due to their relatively long lifetime. In this study, a standardized microwave-excited (ME) APP jet (APPJ) source was employed to produce gaseous RONS to treat liquid samples. The source was a commercially available plasma jet, which generated argon plasma utilizing a coaxial transmission line resonator at the operating frequency of 2.45 GHz. An ultraviolet-visible spectrophotometer was used to measure the concentrations of H_2_O_2_ and NO_3_^−^ in plasma-activated media (PAM). Three different types of media (deionized water, Hank’s balanced salt solution, and cell culture solution Dulbecco’s modified eagles medium [DMEM]) were utilized as liquid samples. Among these media, the plasma-treated DMEM was observed to have the highest levels of H_2_O_2_ and NO_3_^−^. Subsequently, the feasibility of using argon ME-APPJ-activated DMEM (PAM) as an adjuvant to enhance the therapeutic effects of cisplatin on human bladder cancer cells (T-24) was investigated. Various cancer cell lines, including T-24 cells, treated with PAM were observed in vitro for changes in cell viability using the 3-(4,5-dimethylthiazol-2-yl)-2,5-diphenyl-2H-tetrazolium bromide (MTT) assay. A viability reduction was detected in the various cancer cells after incubation in PAM. Furthermore, the study’s results revealed that PAM was effective against cisplatin-resistant T-24 cells in vitro. In addition, a possible connection between HER expression and cell viability was sketched.

## 1. Introduction

Cold atmospheric plasma (CAP) has drawn considerable attention because it contains different reactive and charged species, ultraviolet (UV) light, and electric fields, which can influence biological functions when directly applied to tissue or cells. Consequently, the application of CAP has rapidly expanded to biology, medicine, and agriculture [1,2,3]. Apart from the possibility of directly treating cells or tissues, significant efforts have also been undertaken to utilize the therapeutic capacity of plasma-activated media (PAM) [4,5,6,7,8,9], which can be produced by the CAP treatment of an aqueous solution (e.g., water, phosphate-buffered saline, and other culture media). It is commonly believed that the biochemical activity of PAM is derived from the synergistic effects of highly reactive species, particularly reactive oxygen and nitrogen species (RONS) [9,10]. Recently, CAP and PAM have emerged as novel technologies for cancer therapy.

Different CAP sources have been employed to activate various types of liquids. While CAPs generated by low-frequency driven atmospheric pressure plasma jets (APPJs) or dielectric barrier discharges have been utilized extensively for biomedical applications, microwave-excited (ME) APPJs (ME-APPJs) have several advantages over low-frequency driven plasmas. Microwave-excited plasma differs significantly from the plasmas driven by low- or radio-frequency voltages and has several interesting properties. For example, the electron density is higher in ME compared with radio-frequency or direct current-driven plasmas. Several types of high-density radical species are generated under high electron density circumstances; therefore, the reactivity of ME plasma is expected to be very high [11,12,13].

When operated with ambient air, CAP contains high amounts of RONS, such as ozone (O_3_), nitrogen dioxide (NO_2_) and superoxide (O_2_^−^), hydroxyl (OH), and nitric oxide (NO) radicals. Gaseous RONS produced by CAP can trigger the production of RONS in the CAP-treated liquids. Treatment of liquids with CAP results in activation and non-equilibrium dissociation of water molecules (H_2_O), forming short-lived species, such as hydrogen atoms and OH and NO radicals. Chemical reactions between these species form stable species, such as hydrogen peroxide (H_2_O_2_), nitrite (NO_2_^−^), and nitrate (NO_3_^−^). In the case of indirect PAM treatment, the primary effects are associated with long-lived RONS [9]. The RONS and some of their reaction products exhibit strongly oxidative properties and trigger signaling pathways in living cells [10,13]. Numerous studies have shown that PAM produced by non-equilibrium APPJs is effective against cancer cells in vitro and in vivo [8,9,14,15,16,17,18,19].

Plasma laboratories worldwide use different types of plasma devices, including self-constructed, as they are inexpensive and simple to make. Still, the main drawback of using these homemade devices is that comparison of the results obtained under precisely the same conditions is challenging; this has hampered research efforts and impeded insights into the basic understanding of atmospheric pressure plasmas (APPs) and their interaction with biological tissues and cells, slowing their scaling and preventing official approval for medical applications. To solve this problem, a commercially available jet (MiniJet PM-10.R, Heuermann HF-Technik GmbH, Aachen, Germany) with a microwave signal of about 2.45 GHz was employed in this study; this allowed reproducible experiments helping to standardize the CAP/PAM procedure for cancer therapy.

In previous research, we have explored the feasibility of ME-APPJ as an anti-cancer agent. The PAM was produced using a tuned ME-APPJ and was demonstrated to have a significant cytotoxic effect on A549 cells but little effect on normal cell viability [16]. We observed that A549 cell death was mainly via apoptosis due to a higher level of intracellular ROS and was accompanied by cell cycle arrest, ferroptosis, and caspase-3/7 activation [16,17]. In particular, Jo et al. [17] demonstrated that PAM inhibited tumor growth in a xenograft model with increased lipid peroxidation and decreased ferroptosis suppressor gene expression. However, the cytotoxicity of PAM still needs to be verified for various other cancer cell lines. Furthermore, jet plasma should be characterized to explain the correlation between the plasma properties and RONS generation in the plasma-treated liquid.

During the past decade, possible methods to increase the anti-cancer effect of CAP/PAM in combination with chemotherapy have been explored. Some studies have confirmed added value when combining direct plasma treatment with chemotherapeutic methods [5,20,21,22,23]. For example, Pefani-Antimisiari et al. [24] investigated the combined effects of CAP and the chemotherapeutic drug doxorubicin on normal, and murine and human melanoma cells. Using murine (B16) and human (SK-MEL-28) melanoma cells, Sagwal et al. [23] reported synergistic cytotoxicity of doxorubicin and epirubicin, and additive toxicity of oxaliplatin with direct plasma exposure in drug interaction analyses. The combination treatments led to an increased deoxyribonucleic acid (DNA) damage response. The observed synergistic effects in tumor cells resulted from enhanced intracellular doxorubicin accumulation via upregulation of the organic cationic transporter SLC22A16 by plasma treatment [23]. Conway et al. [22] reported that U373MG glioma cells’ relative resistance to CAP could be overcome when used in combination with low doses of glioblastoma multiforme chemotherapy, temozolomide, or exposure to multiple doses. Another DNA damage-related alkylating agent, cisplatin, has been reported to form crosslinks and damage DNA, activating the DNA damage responses and subsequently inducing apoptosis in cancer cells [25,26]. Li et al. [25] found that PAM highly increased the efficacy of the chemotherapeutic agent cisplatin on hepatocellular carcinoma with cancer stem cell characteristics. Nyguen et al. [27] demonstrated that PAM combined with gemcitabine triggered the deaths of hepatic malignant cells via ROS generation.

Furthermore, Liedtke et al. [5] investigated the combined effect of PAM and cisplatin or gemcitabine in pancreatic cancer cells in vitro. The combination of PAM with drugs, particularly cisplatin, resulted in the death, cycle arrest, and growth inhibition of cells, dramatically enhancing PAM’s anti-cancer effects [5]. Using cells with acquired paclitaxel/cisplatin resistance, Utsumi et al. [28] elucidated the effects of indirect PAM exposure on cell viability and tumor growth in vitro and in vivo. Combined PAM/cisplatin appeared to kill cancer cells more efficiently than PAM or cisplatin alone [28]. The biological anti-cancer mechanism of CAP/PAM requires further investigation regarding the biochemical events and synergistic, or possibly antagonistic effects, with radio-frequency or chemotherapy [29].

This study applied ME-APPJs to various liquid media, anticipating plasma-induced liquid phase chemistry mainly through neutral reactive species. Eliminating any direct effect of ultraviolet (UV) light can be achieved by treating the solution without cells and adding it to the cell medium [4]. The chemical activity of the liquid phase was characterized by detecting the concentration of RONS. The RONS generated in the gas and liquid phases were measured using different diagnostic methods, including optical emission spectroscopy and UV-visible (Vis) spectrophotometry. In addition, the viability of various cancer cell lines, including cisplatin-resistant human bladder cancer cells (T-24), treated with PAM was investigated. Furthermore, a plausible connectivity between cell viability and *HER2/HER3* gene expression was suggested.

## 2. Results

### 2.1. Characterization of the Microwave-Excited Atmospheric Pressure Plasma Jet

The ME-APPJ used for the plasma treatment on the cell culture medium and the schematic diagram of the experimental setup are represented in Figure 1.

Figure 2 presents the measured gas temperature for ME-APPJ as a function of power at three different gas flow rates (1.3, 1.5, and 2.1 standard liters per min [SLM]). The temperature was measured at 2 cm below the jet nozzle. Although the gas temperatures were above room temperature and the treated media evaporated to some extent, there was no serious problem in the treatment of liquids. As the gas flow rate was increased, the gas temperature approached room temperature. 

Figure 3 presents a typical optical emission spectrum ranging from 200–900 nm. The spectrum shows the nitrogen molecule (N_2_) second positive system at 315, 337, 357, and 380 nm, the dinitrogen (N_2_^+^) first negative system at 391 nm, the OH radical band at 309 nm, excited argon (Ar) transition lines (4p → 4s), and excited oxygen (O [^1^D]) bands at 777 nm. Of note are the much higher intensities of the excited OH radical and N_2_ bands. As ME plasma is continuously generated (i.e., not pulsed), many excited particles, such as Ar atoms and electrons, impinge upon the liquid surface. For example, electrons dissociate H_2_O according to the reaction H_2_O + e → •OH + •H + e. The generated OH radicals contribute to the oxidation of organic compounds in the culture medium, while H_2_O_2_ is formed [19].

A relatively high presence of OH, O, and excited N_2_ is attributed to the interaction of ambient air with excited Ar species and high-energy electrons in the plasma. The most predominant chemical species in the plasma jet is metastable state N_2_*(A), which is involved in many chemical reactions leading to the formation of OH, H, H_2_O_2_, NO, NO_2_, nitrous acid (HNO_2_), and nitric acid (HNO_3_) [30]. In contrast to cavity-type-microwave plasma devices operating at medium to high power levels [31], the ME-APPJ utilizing a coaxial transmission line resonator employed in this study had a lower gas temperature and gave rise to a considerable amount of RONS in the gas phase.

### 2.2. Reactive Species in the Plasma-Treated Liquid

#### 2.2.1. Ozone

Ozone (O_3_) plays a role in RONS’s formation in aqueous media, and has been investigated intensely in biomedical applications due to its strong oxidation properties and long lifetime. In addition, O_3_ dissolved in aqueous solutions is an effective antimicrobial oxidant. Ozone is mainly produced via the collision of an oxygen atom (O) and molecule (O_2_) in the gas phase, which then spreads into the liquid phase [32]. The high reaction rate of this process may deplete O [33]. Figure 4 indicates that the O_3_ concentration increased with increasing applied power and decreasing gas flow rate. Since O_3_ is primarily created by a three-body association reaction of O and O_2_, a rise of O with an increase in power can result in increased O_3_ yield [34]. The O_3_ concentration in the gas phase has a tendency to change to that in the liquid phase, which may explain how it is transported into the liquid phase after being produced in the gas phase. Therefore, O_3_ generation becomes slightly lower as the gas flow rate increases.

#### 2.2.2. Electrical Conductivity and pH

Many RONS are produced in a plasma-treated liquid; consequently, the liquid’s physical and chemical properties, including the pH and electrical conductivity, may change markedly. The pH and electrical conductivity are essential in biomedical applications, such as bacterial inactivation [35]. In addition, the biochemical activity of PAM results from the synergistic effects of highly reactive species, particularly the RONS [10].

Figure 5 shows the pH level and electrical conductivity of plasma-treated deionized water (DW), respectively. Comparison is made between three different APPJs: a low-frequency single jet [36], low-frequency jet array [14,37], and ME-APPJ. In addition, Figure 5 shows that the ME-APPJ treatment resulted in more substantial pH and electrical conductivity changes compared with the other APPJs. The decreased pH value suggests a high concentration of RONS in the treated DW, which could explain the increase in the electrical conductivity after the APPJ treatment. The observed changes in the pH and electrical conductivity can be attributed to the concentrations of N_2_*, O, H, and OH radicals generated in the plasma and plasma-liquid transition region. These radicals have been produced by collision between Ar excited (mostly metastable) species and ambient air and the vaporization and dissociation of H_2_O [38].

#### 2.2.3. Hydrogen Peroxide and Nitrate

The ME-APPJ treatment of three different solutions (DW, serum-free Hanks’ balanced salt solution [HBSS], and Dulbecco’s modified eagles medium [DMEM] + 10% fetal bovine serum) were performed to induce the RONS’s changes in the concentration as functions of the operational parameters. To quantify the stable RONS in PAM, H_2_O_2_ and NO_3_^−^ were measured after plasma treatment (3 min) of 3 mL of media using UV-Vis spectrophotometry. Figure 6a shows the change of H_2_O_2_ in the PAM produced at different operating conditions. As shown, H_2_O_2_ increased with increasing power but decreased with increasing gas flow rate (Figure 6a). The presence of OH radicals favors the formation of H_2_O_2_, as the main route of generation in the liquid is the recombination of OH radicals in the gas phase and subsequent diffusion into the liquid phase [38]. As mentioned previously, the main source of OH radicals in the gas phase is the electron impact of H_2_O molecules in the water vapor above and near the water surface [6]. With increasing gas flow rate, the gas (and the electron) temperature decreases; in addition, the production of H_2_O_2_ (OH + OH → H_2_O_2_) and destruction (e + H_2_O_2_ → OH + OH^−^) are enhanced [39]. The net result is decreased H_2_O_2_ with an increased gas flow rate. The H_2_O_2_ concentration in the treated DMEM was observed to be higher than the concentrations in the HBSS and DW. In addition, the DMEM contained a higher concentration of amino acids, vitamins, and additional supplementary components, while the HBSS was composed of various salts only. Furthermore, the plasma might react with amino acids to form NO_2_^−^ components, possibly explaining the higher NO_2_^−^ concentration in the treated DMEM compared with the DW and HBSS [40].

Figure 6b shows the change of NO_3_^−^ in the PAM produced at different operating conditions; the concentration is slightly lower than for the H_2_O_2_. As expected, the NO_3_^−^ levels increased with increasing applied power. When the gas flow rate is increased, the electron temperature decreases; hence, the formation of metastable state N_2_*(A) is oppressed [41]. In addition, NO is formed by various reactions (including N + OH → NO + H; N + O → NO; and N + O_2_ → NO + O), and NO_2_ is formed by reactions such as O_3_ + NO → NO_2_ + O_2_ and 2O + O_2_ → 2NO_2_ [42]. Despite the complex pathways involved, we observed that the production of N_2_*(A) decreased, and the formation of NO_x_ was oppressed with increased gas flow rate [43]. These reactions can result in lower HNO_x_ (the precursor of NO_3_^−^ and NO_2_^−^) in the gas phase, explaining the reduction of the proportionality of NO_3_^−^ change with the gas flow rate compared with those of H_2_O_2_. When the gas flow rate increased from 1.3 to 1.9 SLM, NO_3_^−^ exhibited a decrease in concentration; this can be explained similarly to the H_2_O_2_ case previously described. Hence, the H_2_O_2_ and NO_3_^−^ concentrations increase with increasing applied power while exhibiting a slight decrease with increasing gas flow rate. It is likely that a lower gas flow rate is advantageous for the production of radicals, such as O[^1^D] and nitrogen (N*) atoms, and excited nitrogen (N_2_*), singlet oxygen (O_2_ [^1^Δg]) molecules, negative molecular oxygen ions (O_2_^−^), and OH radicals, which can be utilized for the formation of long-living reactive species [32]. This observation indicated that complex chemical reactions are involved in generating reactive species and the derivatives in the gas and liquid phases, depending on the applied power and gas flow rate.

The high gas temperature obtained by the ME-APPJ provided an environment for nitrogen dissociation via the generation of N_2_ molecules. In addition, ME plasma can achieve a high electron density and temperature, which could help produce abundant Ar(3p) species. The high-energy components of the plasma are responsible for creating reactive species, including RONS. In an Ar ME-APPJ, the flux of highly reactive or short-lived species, such as Ar metastable, OH radicals, and atomic oxygen (O_1_), which give rise to further RONS, should be considered [44]. The N_2_ and O_2_ molecules in air dissociate to form NO and NO_2_, which may be converted into acids, such as HNO_2_ and HNO_3_ [42,45].

### 2.3. Cell Viability Induced by Plasma-Activated Media

The RONS in PAM contribute to cellular oxidative stress, which leads to cell death [5,6,7,8]. The cell viability was checked using the 3-(4,5-dimethylthiazol-2-yl)-2,5-diphenyl-2H-tetrazolium bromide (MTT) assay to test the hypothesis that PAM could induce cell death. The cytotoxic effect of PAM on various cancer cell lines, including human colon cancer cell lines (HCT116, RKO, HT29, and DLD-1), bladder cancer cell line (T-24), and human breast cancer cell line (MDA-MB-231), was tested. Figure 7 clearly shows PAM’s “dose” effect—with increasing plasma exposure time, the decrease of cell viability was apparent in all cancer cell types tested.

Furthermore, the effect of PAM on the viability of various cancer cell lines was evaluated at 24, 48, and 72 h post-PAM treatment (i.e., PAM incubation time). Figure 8 shows that cell viability decreased with increasing incubation time. Notably, the T-24 and RKO cancer cells data at 24 and 48 h post-PAM application showed that the longer the cells were exposed to PAM, the stronger the effect on cell viability; more than 50% of the viable cells were killed at 24 h post-PAM application [46].

### 2.4. Synergistic Effects of Combined Plasma-Activated Media and Cisplatin on Cancer Cells

We hypothesized that plasma treatments may enhance the cellular uptake of chemotherapeutic drugs. The cell viability for PAM or cisplatin alone and combined PAM and cisplatin treatments were determined to confirm if the lethality of combined PAM and cisplatin was synergetic. Moreover, to test the potential of PAM to enhance cellular uptake of chemotherapeutics in chemo-resistant cancer cells, T-24 and cisplatin-resistant T-24 (T-24R) cells were treated with cisplatin in PAM that had been freshly irradiated by plasma.

Figure 9 shows that the PAM-alone treatments (30P [plasma exposure time 30 s] and 60P [exposure time 60 s]) were more lethal to T-24 cells compared with the 5 µM of cisplatin-only treatment. However, the combined PAM and cisplatin (30P+C and 60P+C) treatments improved the therapeutic effects and resulted in lower viability in T-24 and T-24R cells compared with the PAM-alone and cisplatin-alone treatments. In particular, the PAM-alone treatment (30P) was ineffective in reducing T-24R cells’ viability; there was no significant difference in cell viability between the PAM-alone (60P) and 25 µM of cisplatin-alone treatments. The results indicated that PAM facilitated cell deaths by substantially sensitizing malignant cells to cisplatin, and that a higher dose of PAM is needed to reduce cell viability in T-24R cells.

## 3. Discussion

Human epidermal growth factor receptor 3 (HER3, ErbB3), a member of the epidermal growth factor erythroblastic leukemia viral oncogene homolog (ErbB)/HER receptor tyrosine kinases family, has been shown to play a crucial role in the regulation of oncogenic processes [47] and is known to express in various types of tumors. Figure 8 shows that colorectal cancer cell lines demonstrated different sensitivities to PAM. The four colon cancer cell lines used in the experiment can be divided into two groups based on the presence of HER3; HT29 and DLD-1 cells are considered HER3 positive, while RKO and HCT116 cells are HER3 negative [47]. Previous studies have indicated that HER3 is a protective factor in the cell death pathway activated by oxidative stress [48]. In contrast, it has been reported that the expression of HER2 (ErbB2) and HER3 is induced when ovarian cancer cells are exposed to reactive oxygen species (ROS) [49].

Further, ErbB2 is a member of the membrane-spanning type I receptor tyrosine kinase family, comprising four closely related family members [50]. Overexpression of ErbB2 hyperactivates components of the cell cycle machinery and is linked to resistance against apoptosis-inducing therapeutic agents. Mujoo et al. [51] demonstrated that HER3 knockdown induces cell cycle arrest and apoptosis of colon cancer cell lines by activating Bcl-2-associated X protein (Bax) and Bcl-2 antagonist killer 1 (Bak). These studies alluded to the critical involvement of HER3 in colon cancer progression [47]. Accumulating evidence supports the role of HER3 in developing resistance to epidermal growth factor receptor-targeted therapies [52].

To investigate whether the expression of HER2/HER3 is related to the induction of ROS by PAM in colon cancer cells, ribonucleic acid (RNA) expression changes were examined using quantitative real-time polymerase chain reaction (qRT-PCR). The results indicated a considerable change in expression in all the cancer cell lines (Figure 10). Notably, the changes in HER2 and HER3 expression in the RKO cell line, which belongs to the HER3 negative group, were significant.

In Figure 7, we confirmed the effect of PAM on four colon cancer cells. The result indicates that the effect of PAM varies depending on the cell line. We sought to find the cause of these differences. The four cell lines could be divided into two groups according to the presence or absence of HER3. However, contrary to our expectations, the differences in cell viability were not in accordance with the differences in mRNA levels. The mRNA level measurement indicated that HER2/3 gene expression was greatly changed in RKO cells. From this, it may be inferred that differences in cell viability within colon cancer cells may not be related to the presence or absence of HER3. The reason for a significant increase in HER2/3 gene expression by PAM in RKO cells is not clear at the moment. There may be a connection between HER expression and cell viability because an appreciable decrease in cell viability was observed in the RKO cell line. However, their connectivity on the molecular level should be explored further in future studies.

Previous studies have reported that intracellularly accumulated ROS critically contributes to the decline in mitochondrial membrane function [9,53]. In addition, PAM can induce mitochondrial damage under experimental conditions because the generation of ROS was detected by the PAM treatment [16,17]. Adachi et al. [9] investigated the expression levels of the mitochondrial antiapoptotic proteins B-cell lymphoma 2 (Bcl2) and proapoptotic Bax to elucidate the mechanisms underlying PAM-induced mitochondrial injury; the PAM suppressed the expression of Bcl2 but not Bax. Moreover, mitochondrial dysfunction and endoplasmic reticulum stress interact to disrupt each other and facilitate cellular injury. In addition, the PAM and chemotherapeutics showed synergetic therapeutic effects. A possible explanation for the enhancement of cisplatin uptake is lipid oxidation caused by oxidative stress from ROS, which is further attributed to pore formation [54] or to an increase in the expression of aquaporin 5 [55]. Hence, enhancing molecule uptake by cells after PAM treatment might facilitate the synergistic lethal effect of PAM and chemotherapeutics on cancer cells [56].

## 4. Materials and Methods

### 4.1. Plasma Jet and Diagnostics

A commercially available MiniJet PM-10.R utilizing a coaxial transmission line resonator at 2.45 GHz resonance frequency was employed. The details of this APPJ can be found in our earlier study [16]. The MiniJet was controlled with input power between 5–8 W and gas flow rates of 1.3–2.1 SLM.

To identify the gaseous RONS in plasma, optical emission spectroscopy was used to detect NO, OH, N_2_, N_2_^+^, O, H, and Ar bands. The optical emission spectrum of the APPJ was measured using a SPEX 1702 spectrophotometer monochromator (Spex Industries, Inc., Metuchen, NJ, USA) with a KR928 photomultiplier tube (Hamamatsu Photonics K.K., Honshu, Japan). The gas temperature *T*_G_ was measured using a Luxtron^®^ fiber-optic temperature sensor (M601-DM&STF; Advanced Energy Industries, Inc., Denver, CO, USA) [32].

Ozone is an important reactive species formed by atmospheric pressure plasma. Therefore, O_3_ generation was detected using an O_3_ detector (Model 202 Ozone Monitor; 2B Technologies, Broomfield, CO, USA) based on UV light’s absorption at 254 nm [32]. Several authors have described the role of NO and related compounds as a direct cancer therapy or an agent that re-sensitizes tumors to chemotherapy or radiation therapy [53,57,58]. These studies demonstrated an important tumor apoptotic intercellular signaling mechanism involving NO and peroxynitrite (ONOO^−^) [36].

### 4.2. Measurement of Reactive Oxygen and Nitrogen Species Concentration in the Plasma-Treated Medium

Plasma generated from atmospheric air contains chemically reactive H, N, O, O_3_, NO, NO_2_, OH, and H_2_O_2_ [32]. When a solution is exposed to plasma, some of these RONS dissolve from the gas phase into the solution. The final products of these unstable species in the liquid phase might be O_2_ (^1^Δg), H_2_O_2_, O_3_, NO_x_, N_2_O, and HNO_x_, which are relatively stable and interact with cells resulting in the production of intracellular RONS [32,53]. In addition, H_2_O_2_, HNO_2_, and HNO_3_ have high Henry’s law solubility constants; thus, they are continuously transported from the gas phase into the liquid during plasma treatment [59]. In addition to the accumulation of H_2_O_2_ in the liquid because of transport from the gas phase, there is a net production of H_2_O_2_ in the liquid phase. The dominant mechanism of H_2_O_2_ production is the recombination of two OH radicals, which occurs primarily in the first few nm underneath the gas–liquid interface and also in the center of the well (i.e., the depression of the dimple) because further and deeper into the liquid there are few OH radicals left. In the same region, the loss of H_2_O_2_ is mainly caused by OH radicals (i.e., H_2_O_2_ + OH → HO_2_ + H_2_O), while deeper in the bulk liquid, the H_2_O_2_ loss is caused by a reaction with H radicals (i.e., H_2_O_2_ + H → OH + H_2_O). As previously mentioned, the short-lived species (O, OH, NO, NO_2_, and NO_3_) immediately react in the interface region to form other (long-lived) species; in this way, they cannot reach the bulk liquid and, hence, do not accumulate in the solution. Only H_2_O_2_, HNO_2_/NO_2_^−^, and HNO_3_/ NO_3_^−^ are stable in the buffered solution after plasma treatment [59].

A photoLab^®^ 7600 UV-VIS spectrophotometer (Xylem Analytics Germany Sales GmbH & Co. KG WTW, Weilheim, Germany) was used to measure the concentrations of H_2_O_2_, NO_3_^−^, and O_3_ in plasma-treated liquids. The concentration levels of H_2_O_2_ and NO_3_^−^ in the plasma-treated liquids were investigated as functions of the applied power and gas flow rate. Three solutions (DW, HBSS, and DMEM) were employed as liquid samples. The DW’s electrical conductivity and pH were measured using a conductivity stick meter (Starter ST3100C) and a digital pH meter (Starter ST3100) both Ohaus Corp., Parsippany, NJ, USA. Although the concentration of the active species is the highest at the tip of the plasma plume [60], the treated liquid surface was placed at 5 mm below the plume tip to prevent severe evaporation.

### 4.3. Cell Culture Cell Viability, and Quantitative Real-Time Polymerase Chain Reaction Analysis

#### 4.3.1. Cell Culture

The T-24, HT29, and DLD-1 cells were grown in DMEM (Capricorn Scientific GmbH, Ebsdorfergrund, Germany) and the MDA-MB-231, RKO, and HCT116 cells were incubated in Roswell Park Memorial Institute with L-glutamine media (Capricorn Scientific GmbH, Ebsdorfergrund, Germany). The media were supplemented with 10% fetal bovine serum and 1% penicillin/streptomycin (Capricorn Scientific GmbH, Ebsdorfergrund, Germany). The T-24R cell line was provided by Professor Sun-Hee Leem (Dong-A University, Busan, Republic of Korea). The cells were incubated in a humidified atmosphere with 5% CO_2_ at 37 °C.

#### 4.3.2. Cell Viability

Cell viability was measured using the MTT assay (Duchefa Farma, Haarlem, The Netherlands). Cells were seeded in a 96-well plate at 1 × 10^4^ cells/well. The PAM was removed after incubation, replaced with 100 µL of MTT solution (5 mg/mL), and incubated for 3 h. The violet crystals were solubilized with dimethyl sulfoxide, and the absorbance was measured at 550 nm using a microplate reader (FLUO star OPTIMA; BMG Labtech, Ortenberg, Baden-Wrttemberg, Germany). The cell viability (%) was calculated as follows: optical density of treated cells/optical density of non-treated cells × 100.

#### 4.3.3. Quantitative Real-Time Polymerase Chain Reaction Analysis

Total RNA was extracted with an AccuPrep^®^ Universal RNA Extraction Kit (Bioneer Corp., Daejeon, Republic of Korea), and complementary DNA (cDNA) was synthesized with PrimeScript RT Master Mix (Takara Bio, Shiga, Japan), following the manufacturer’s protocol. Then, qRT-PCR was carried out using TB Green^®^ and QuantStudio 3 (Thermo Fisher Scientific Inc., Waltham, MA, USA). The primers were purchased from Bioneer Corp.

#### 4.3.4. Statistical Analysis

All data are reported as mean ± standard deviation (SD). Statistical significance of the difference between groups was analyzed by a two-tailed unpaired or paired Student’s *t*-test or by one-way analysis of variance (ANOVA) with Tukey’s post hoc test in instances of multiple comparisons (Prism, version 9; GraphPad Software, Inc., San Diego, CA, USA).

## 5. Conclusions

An Ar ME-APPJ was used to prepare PAM. Although the gas temperature of the ME-APPJ was high, its application to PAM was appropriate, practical, and helpful, especially for cancer cell treatment. The diversity of the antitumor efficacy of the PAM differed depending on the cell type and culture medium; therefore, it is essential to understand how these differences affect the understanding and successful application of PAM. Therefore, the properties of PAM produced by ME-APPJ were explored regarding the chemical composition of long-lived reactive species and its antitumor activity. Among the culture media tested, the plasma-treated DMEM was observed to have the highest levels of H_2_O_2_ and NO_3_^−^. Furthermore, it was demonstrated that PAM inhibited cell growth in various cancer cell lines. The effects of the PAM alone and combined with cisplatin on treating T-24 and T-24R cells were examined. Combined treatment of PAM and cisplatin was demonstrated to act synergistically against cancer cell growth in vitro. Additionally, a significant increase in HER2/3 gene expression by PAM was observed in human colon cancer cells. The relationship of HER2/3 gene expression with cell viability should be clarified in future studies.

## Figures and Tables

**Figure 1 ijms-25-01249-f001:**
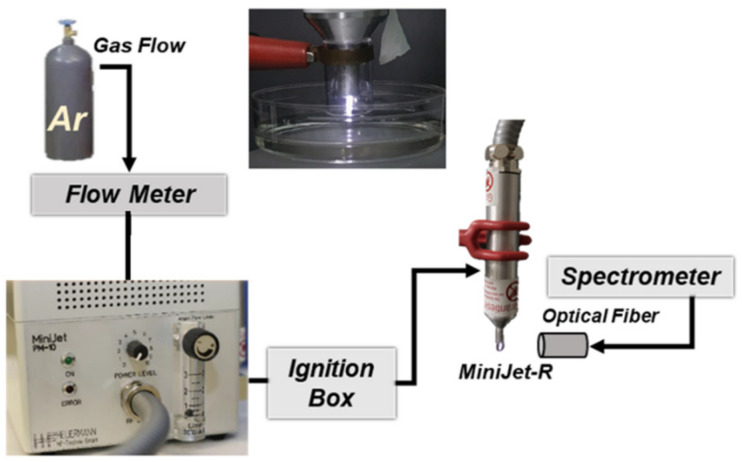
Photograph of the plasma plumes from a microwave-excited atmospheric pressure plasma jet (ME-APPJ) contacting a medium in a liquid container, and schematic diagram of the experimental setup.

**Figure 2 ijms-25-01249-f002:**
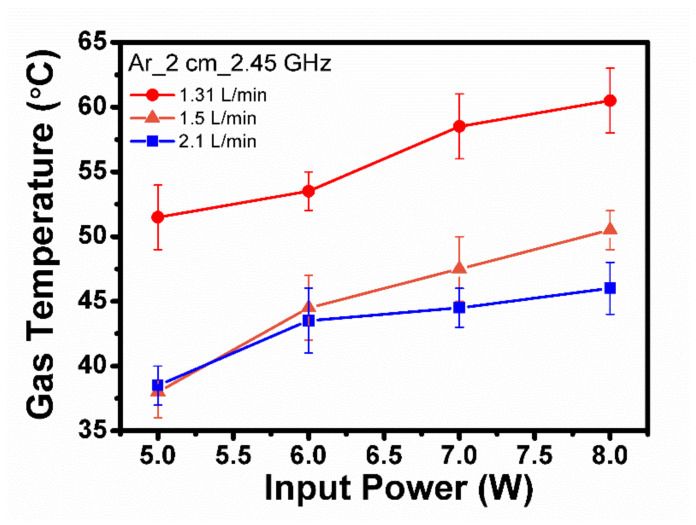
Gas temperature for the microwave-excited atmospheric pressure plasma jet (ME-APPJ) as a function of power at three different gas flow rates (1.3, 1.5, and 2.1 SLM). Each point represents the mean ± SD of three replicates.

**Figure 3 ijms-25-01249-f003:**
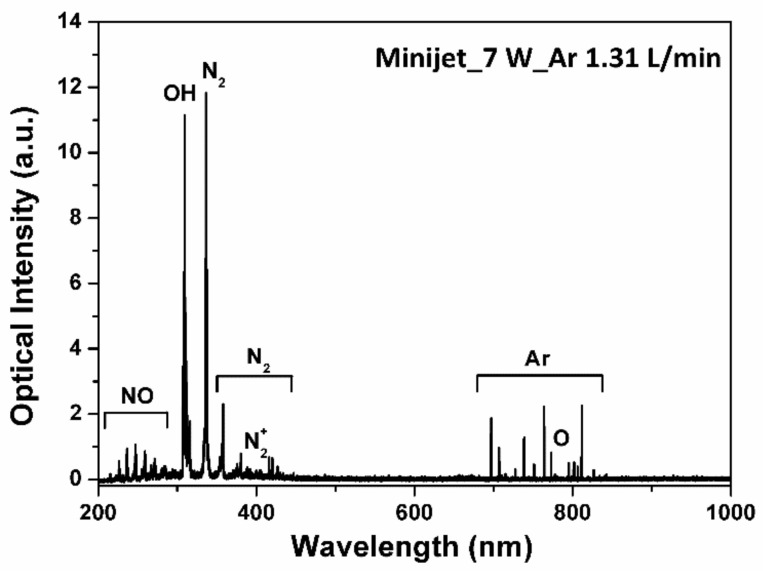
Typical optical emission spectrum for microwave-excited atmospheric pressure plasma jet (ME-APPJ). The power is 7 W and the gas flow rate is 1.3 SLM.

**Figure 4 ijms-25-01249-f004:**
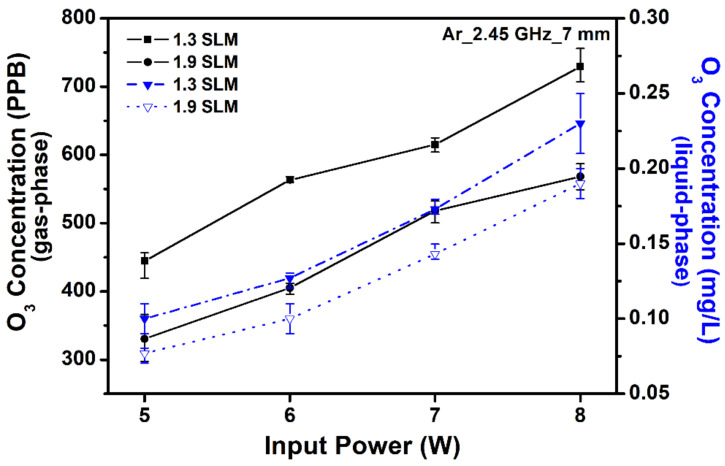
Ozone concentration in the gas phase and plasma-activated media (PAM) as a function of power at different gas flow rates (1.3 and 1.9 SLM). Each point represents the mean ± SD of three replicates.

**Figure 5 ijms-25-01249-f005:**
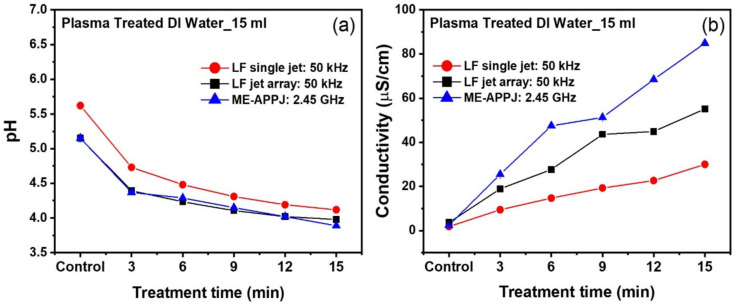
(**a**) The pH and (**b**) electrical conductivity of the plasma-treated distilled water (DW). Comparison between three different atmospheric pressure plasma jets (APPJs): low-frequency single jet [36], low-frequency jet array [14,37], and microwave-excited (ME)-APPJ. Each point represents the mean ± SD of three replicates.

**Figure 6 ijms-25-01249-f006:**
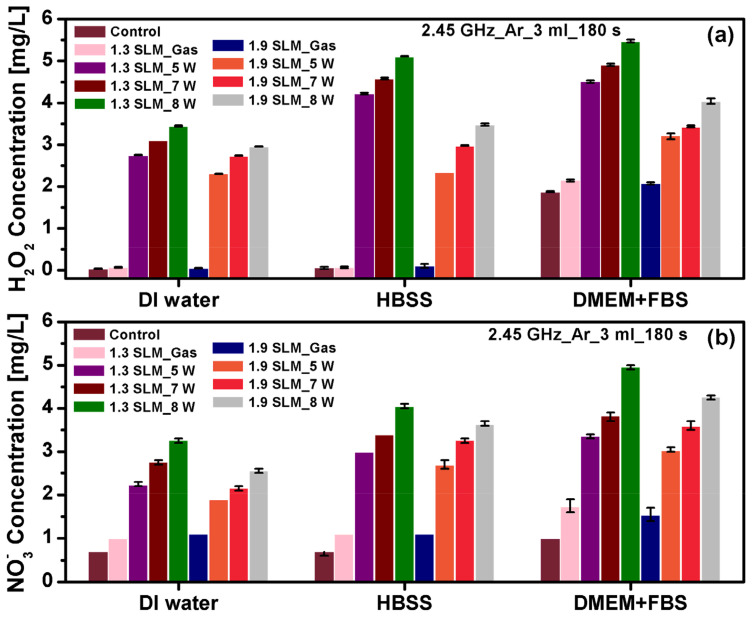
Concentrations of (**a**) H_2_O_2_ and (**b**) NO_3_^−^ produced in the plasma-treated media (deionized water [DW], serum-free Hanks’ balanced salt solution [HBSS], and Dulbecco’s modified eagles medium [DMEM]) at different conditions. The concentration of H_2_O_2_ and NO_3_^−^ was detected as a function of the power at different gas flow rates. Each point represents the mean ± SD of three replicates.

**Figure 7 ijms-25-01249-f007:**
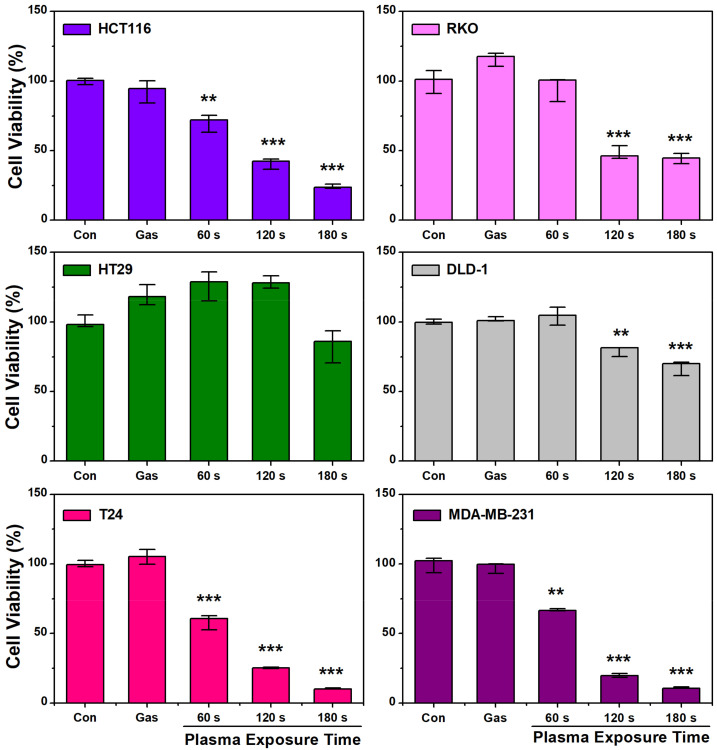
Cell viability of various cancer cell lines (human colorectal cancer cell line [HCT116, RKO, HT29, and DLD-1], human bladder cancer cell line [T-24], and human breast cancer cell line [MDA-MB-231]), detected using a 3-(4,5-dimethylthiazol-2-yl)-2,5-diphenyl-2H-tetrazolium bromide (MTT) assay kit. Data are represented as the mean ± SD. Cell viability was measured 24 h after the plasma-activated media (PAM) treatment. The jet operating conditions were a gas flow rate of 1.9 standard liters per min (SLM) and power of 7 W. The culture medium was exposed for 60, 120, and 180 s. **, *p* < 0.01; ***, *p* < 0.001.

**Figure 8 ijms-25-01249-f008:**
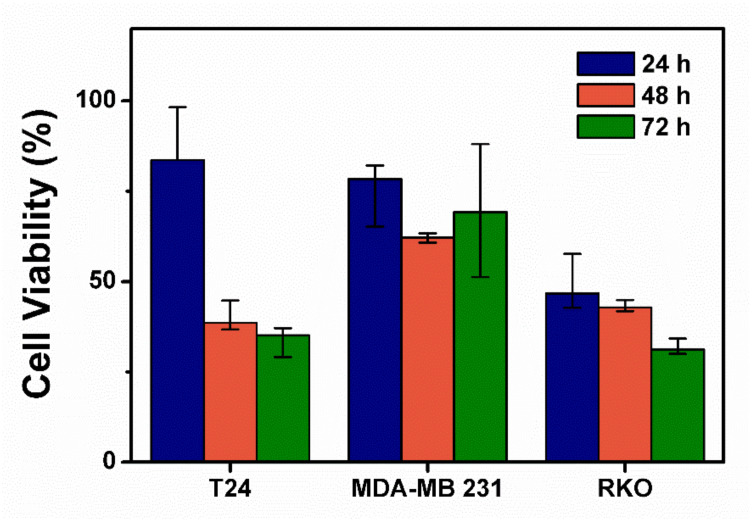
Cell viability of various cancer cell lines (bladder cancer cell line [T-24], human breast cancer cell line [MDA-MB-231], human colorectal cancer cell line [RKO]) at several incubation times (24, 48, and 72 h). Cells were treated by plasma-activated media (PAM) produced by a plasma jet with the operating conditions: 1.9 SLM, 7 W, 120 s. Each point represents the mean ± SD of three replicates.

**Figure 9 ijms-25-01249-f009:**
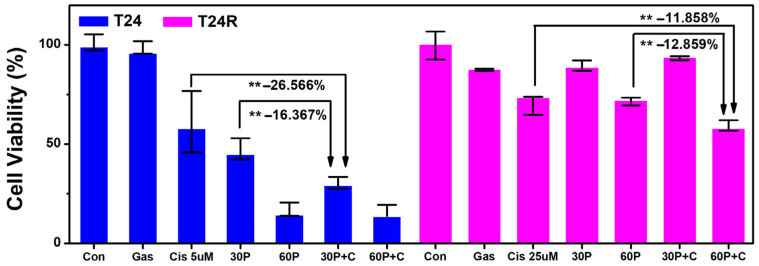
Plasma-activated media (PAM) sensitizing human bladder cancer cells (T-24) to cisplatin. T-24 and cisplatin-resistant human bladder cancer cells (T-24R) were treated with PAM for 24 h. For the T-24 cells, the PAM-alone treatments 30P (plasma exposure time 30 s) and 60P (exposure time 60 s) proved to be effective relative to the cisplatin (5 µM)-only treatment. For the T-24 cells and T-24R, the combination of the PAM with cisplatin (30P+C and 60P+C) resulted in lower viability as compared to the PAM-alone case and the cisplatin-alone (25 µM) case. Each point represents the mean ± SD of three replicates. **, *p* < 0.01. The percentage values shown along with ** represent the reduction of the measured viability.

**Figure 10 ijms-25-01249-f010:**
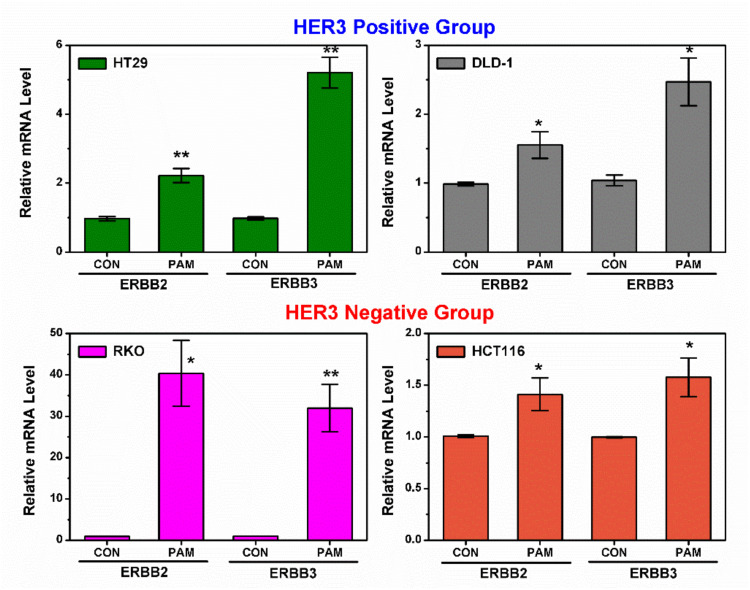
Plasma-activated media (PAM) regulated mRNA expression levels of HER2 (ErbB2) and HER3 (ErbB3) genes in HT29, DLD-1, RKO, and HCT116 cancer cells measured by quantitative real-time polymerase chain reaction (qRT-PCR). Cells were treated by plasma-activated media (PAM) produced by a plasma jet with the operating conditions: 1.9 SLM, 7 W, 180 s. Each point represents the mean ± SD of three replicates. *, *p* < 0.05; **, *p* < 0.01.

## Data Availability

All data are contained within the manuscript.

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
