# Peer review of "Plasma-Activated Media Produced by a Microwave-Excited Atmospheric Pressure Plasma Jet Is Effective against Cisplatin-Resistant Human Bladder Cancer Cells In Vitro"

_ijms, 2024, doi:10.3390/ijms25021249_

Round 1
Reviewer 1 Report
Comments and Suggestions for Authors
COMMENTS TO THE AUTHOR(S)
The study aims to characterize Microwave-excited plasma jet and its effect on Cisplatin-resistant human bladder cancer cells. The study is novel, however, some important sections of the manuscript are error or missed, and I have the following comments for further improvement of the manuscript to be suitable for publication:
Results
2.1 characterization of the microwave-excited atmospheric pressure plasma jet
- Does the plasma effluent length of the MiniJet-R remain unchanged as the input power increases?
- line 16 at page 4, “Both spectra show the nitrogen molecule second positive system at ~~~, “, there are another spectrum exist in Fig 3? If only one spectrum, the word “Both” need to be change.
- line 19 at page 4, “excited argon (Ar) and excited oxygen (O[1D]) bands at 777 nm. -> the spectrum of excited argon is not 777 nm.
- line 29 at page 4, “(HNO3]” bracket errortypo.
- line 29 at page 4, “In contrast to cavity-type-microwave plasma devices,~~ the ME-APPJ utilizing a coaxial ~~ gave rise to a considerable amount of RONS in the gas phase. -> In literatures [31], there are no mention and data about gas temperature and RONS in the gas phase for comparation. On what basis do the authors claim that ME-APPJ has lower gas temperature and considerable amount of RONS?
- what is the reason of the chosen gas flow rate (1.3, 1.5, 2.1 LPM)?
2.2 Reactive species in the plasma-related liquid
2.2.1 ozone
- In Figure 5, why are the units for the left and right axes different? To better understand the sentence “The O3 concentration in the gas phase has a tendency to change to that in the liquid phase”, it is better to specified y-axis title to distinguish in gas phase or liquid phase.
- line 17 at page 5, “Therefore, O3 generation ~” why O3 generation becomes lower as gas flow rate increases?
2.2.2 Electrical conductivity and pH
- In figures 5(a) and 5(b), I recommend that addition of control data (ex: pH of non-treated DIwater and conductivity of non-treated DIwater).
2.2.3 Hydrogen peroxide and nitrate
- line 25 at page 6, “These radicals form because of plasma expansion~”, plasma expansion is not appropriate word for generation of radicals.; Author used Ar ME-APPJ, then the expression is “these radicals has been produced by collision between Ar excited (mostly metastable) species and ambient air ~ ” better to understand for reader.
- line 40 at page 6, “the gas (and the electron) temperature decreases; in addition ~~”, why destruction (e+H2O2->OH+OH-) reactions enhanced as electron temperature decreased.?
- line 19 at page 7, “When the gas flow rate increases, the electron temperature decreases~ [41]”, the reference [41] was not mentioned electron temperature decreasing by flow rate, they mentioned gas temperature decreasing. And also, reference [41], they used mixed gas and increase of ratio of N2 or O2 flow rate in the mixed gas, not total flow rate.
2.3 cell viability induced by plasma-activated media
- Figure 8 at page 10, what exact PAM condition used? (plasma exposure time 60 s ,120 s, 180 s?)
2.4 Synergistic effects of combined plasma-activated media and cisplatin on cancer cells.
- Figure 9 at page 10, some sentence is not appropriate in the figure caption. (ex: however, combined treatment improvement~). these sentences should be move to section 2.4 for description of Figure 9.2.
3. Discussion
- Figure 10 at page 11, all mRNA level of HER2 and HER3 by PAM show higher than control. HCT116 shows relatively lower increase than control, but cell viability in Fig 7 is highly decreased. why it is different with the RKO case?
Comments on the Quality of English Language
Minor editing of English language required
Reviewer 2 Report
Comments and Suggestions for Authors
It's a good work and I believe it should be published. Just a few considerations.
The MTT assay is an assessment of metabolism and not cell viability. The annexin V assay, which the authors carried out in another work of theirs that they cite in reference 16, is what makes it possible to evaluate cell viability and be able to reach this conclusion with the work.
The conditions for treating the medium with CAP should be made more explicit, as variables such as the contact area and the volume of liquid treated have been shown in previous works to influence the amount of RONS obtained.
The choice of mediums to treat with CAP must also be justified. Although DMEM makes sense from a laboratory perspective, it has no clinical translation. Mediums such as saline solution would be more interesting for both points of view.
Only studies of RONS content were carried out, but it would also have been interesting to evaluate anti-oxidant defenses to understand the cell's response to oxidative insult.
The study of HER3 expression and its relationship with PAM exposure could be considered for a separate publication. It does not contribute to the main objective of the work (the synergistic action between cisplatin and PAM in the treatment of bladder tumor), it is not even related to the bladder tumor, it is not mentioned in the abstract or in the keywords. It will be a difficult to find result for other researchers to contribute to the development of scientific knowledge.
And the discussion should be further developed regarding the main result of the work. Cisplatin is an alkylating agent and RONS can themselves induce DNA breaks. Some studies show a relationship between the effects of cisplatin and the expression of aquaporin 5 and others show how PAMs increase the expression of aquaporin 5. These are 2 examples that could enrich the discussion.
